# Association between serum uric acid, hyperuricemia and low muscle mass in middle-aged and elderly adults: A national health and nutrition examination study

**Laixi Kong**[1], **Yaqin Li**[2,3], **Rong Zhu**[4,5], **Maoting Guo**[1], **Yuqing Wu**[1], **Yuxin Zhong**[1], **Zhe Li**[6,7]*, **Zhenzhen Xiong**[1]*

1 School of Nursing, Chengdu Medical College, Chengdu, Sichuan, China, 2 School of Nursing, The Hong Kong Polytechnic University, Chengdu, Sichuan, China, 3 School of Nursing, West China Hospital, Sichuan University, Chengdu, Sichuan, China, 4 The 3rd Affiliated Hospital Of Chengdu Medical College, Chengdu, Sichuan, China, 5 Pidu District People's Hospital, Chengdu, Sichuan, China, 6 Sichuan Clinical Medical Research Center for Mental Disorders, Chengdu, Sichuan, China, 7 Mental Health Center, West China Hospital, Sichuan University, Chengdu, Sichuan, China

* xzz62308631@163.com (ZX); jay_li@163.com (ZL)

**Data Availability Statement:** The data underlying the results presented in the study are from the National Health and Nutrition Examination Survey

## Abstract

### Backgrounds

Recent research suggests that uric acid, as a metabolite with antioxidant properties, may affect muscle function and health. However, the association between serum uric acid (SUA) and low muscle mass remains relatively obscure. This study focuses on the association between SUA and low muscle mass in a middle-aged and elderly population in the United States.

### Methods

Utilizing data from the National Health and Nutrition Examination Survey (NHANES), a total of 12,106 patients aged ≥45 years, possessing complete analytical data, were incorporated. Low muscle mass in our study is defined as indices below 0.789 for males and 0.512 for females, according to the FNIH Biomarkers Consortium. Gender stratified analyses were conducted employing a multivariate weighted logistic regression model. When examining serum uric acid (SUA) levels, the SUA dataset was stratified into deciles, and odds ratios (ORs) were calculated across distinct subgroups of males and females. A restricted cubic spline (RCS) method was employed to investigate the potential nonlinear association between SUA levels and low muscle mass. A series of subgroup analyses stratified by demographic variables and clinical experience were conducted.

### Results

A total of 2,185 participants (18.05%) were identified with low muscle mass, comprising 1,121 males and 1,064 females. Females with low muscle mass had higher SUA levels and an increased incidence of hyperuricemia compared to those without low muscle mass. In

(NHANES).The survey data are publicly available on the internet for data users and researchers throughout the world (www.cdc.gov/nchs/nhanes/).(accessed on 10 October 2023).

**Funding:** The funding for this study was provided by Special Project for Strategic Cooperation between Sichuan University and Dazhou Municipal People's Government (2022CDDZ-17, awarded to ZX) and the 2023 Clinical Science Research Fund Project of Chengdu Medical College, the Third Affiliated Hospital of Chengdu Medical College and Chengdu Pidu District People's Hospital, along with the Open Project of the Sichuan Collaborative Innovation Center for Aging and Elderly Health (23LHPDZYB24, awarded to ZX). The funders had no role in study design, data collection and analysis, decision to publish, or preparation of the manuscript.

**Competing interests:** The authors have declared that no competing interests exist.

females, a fully adjusted multivariable weighted regression model revealed a positive association between hyperuricemia and low muscle mass (OR, 1.43; 95% CI, 1.06 to 1.92; P = 0.021). No significant association was observed in males. Additionally, RCS curves indicated a J-shaped relationship between increasing SUA levels and the risk of low muscle mass in females, and an inverse J-shaped relationship in males.

## Conclusions

This study reveals a significant positive correlation between hyperuricemia and the risk of low muscle mass in middle-aged and older women in the United States, whereas the relationship between SUA levels and low muscle mass did not attain statistical significance. In the male cohort, neither SUA levels nor hyperuricemia demonstrated a significant association with low muscle mass.

## Introduction

Sarcopenia is a pathological condition characterized by the progressive diminution of muscle mass and functionality as the aging process unfolds [1]. The prevalence of global sarcopenia ranges from 10% to 27% among individuals aged over 60 years. Disparities in prevalence rates are attributable to variations in diagnostic methodologies and demographic factors across different regions [2]. Apart from predominantly manifesting in the elderly population, the decline in muscle mass may also be observed in individuals entering middle age, commencing around the age of 40 [3]. Although sarcopenia is associated with age, it is not solely a foregone conclusion of the intrinsic aging trajectory. Its development is susceptible to an array of influencing factors, including poor nutrition, lack of exercise [4], chronic disease [5, 6], medication side effects [7]. Low muscle mass, a critical characteristic of sarcopenia, is particularly vulnerable to deterioration due to endocrine alterations linked with aging [8]. Nevertheless, the association between serum uric acid levels and low muscle mass remains comparatively ambiguous.

Serum uric acid (SUA), the final product of purine metabolism endowed with antioxidant capabilities, primarily emanates from the enzymatic breakdown of nucleic acids and other purine compounds from cellular metabolism, as well as purines present in food [9, 10]. Hyperuricemia, on the other hand, ensues from disruptions in purine metabolism, resulting in a continual elevation of uric acid levels, ultimately progressing into a state of hyperuricemia. Findings from a cross-sectional investigation in the United States have indicated a prevalence of approximately 20% for hyperuricemia, thus emerging as a significant public health concern [11]. Moreover, serum uric acid levels are intricately linked to various health outcomes, including congestive heart failure, hypertension, impaired fasting glucose or diabetes in the general population, chronic kidney disease, and cardiovascular disorders [12, 13].

Previous investigations have revealed intricate interactions between SUA and muscle mass, strength, and functionality. A cross-sectional survey involving adult males in Japan demonstrated an inverted J-shaped relationship between SUA levels and muscle mass [14]. Additionally, a study conducted in the United States specifically utilized the skeletal muscle mass index to investigate the relationship between uric acid levels and muscle quality, revealing a negative correlation between uric acid levels and muscle mass [15]. Furthermore, as research in this field continued to delve deeper, it was discovered that SUA might confer a protective effect against age-related muscle strength decline in a community-based cohort of elderly Japanese

 

women [16]. This outcome aligns with similar findings in studies conducted in elderly Chinese populations. It was observed that there are disparities in the results between male and female cohorts, although the exact underlying mechanism for these differences remains undetermined [17]. Recent cross-sectional and cohort studies in China employing the skeletal muscle mass index (SMI) have consistently demonstrated an inverse correlation between SUA and sarcopenia [18, 19]. Conversely, in both a Brazilian kidney transplant population and an elderly Italian cohort, SUA exhibited a positive correlation with muscle mass [20, 21]. It may be due to disparities in East-West lifestyles. Nevertheless, research elucidating the linkage between SUA and low muscle mass remains sparse and inconclusive. Considering the elevated prevalence of hyperuricemia and sarcopenia in the United States, we leveraged the National Health and Nutrition Examination Survey (NHANES) database for a comprehensive clinical cross-sectional analysis spanning the periods 1999–2006 and 2011–2018. Our study aimed to elucidate the associations between SUA, hyperuricemia, and low muscle mass status, as determined by dual-energy X-ray absorptiometry, among the middle-aged and elderly U.S. population.

## Methods and study design

### Study population

The data for this study were derived from the NHANES, sponsored by the National Center for Health Statistics (NCHS). The NCHS is a component of the Centers for Disease Control and Prevention (CDC), responsible for furnishing crucial health statistical data for the nation. NHANES collects demographic and comprehensive health information through home visits, screenings, and laboratory tests conducted by Mobile Examination Centers (MEC). The survey comprises cross-sectional interview, examination, and laboratory data, collected from a complex multistage, stratified, clustered probability sample representative of the U.S. population. Ethical approval for NHANES is granted by the Institutional Review Board of the National Center for Health Statistics (IRBNCHS), and all participants provided written informed consent before participation. This study is based on retrospective analysis and lacks personally identifiable information. Secondary analyses did not necessitate additional institutional review board approval.

Due to the absence of skeletal muscle mass measurement records for the years 2007–2010, the study scope was confined to individuals who participated in the surveys during the periods of 1999–2006 and 2011–2018 [22]. Among the NHANES participants from 1999–2006 to 2011–2018, a total of 80,630 individuals were included in the investigation. Exclusion criteria were as follows: (a) participants aged below 45 (n = 56183), (b) participants lacking data on appendicular skeletal muscle mass, body mass index, energy and protein intake, and SUA, and (c) participants with missing age or gender information. In instances where covariate missing values remained below 10%, mean multiple imputation was employed to minimize potential biases. Ultimately, this cross-sectional study comprised 11,784 participants analyzed (Fig 1).

### Exposure variable

The primary exposure variable in this study are SUA and hyperuricemia. SUA levels were measured using Beckman UniCel® DxC800 Synchron or Beckman Synchron LX20 (Beckman Coulter, Inc., Brea, CA, United States) at Collaborative Laboratory Services. Following established diagnostic criteria, hyperuricemia in this study was defined as levels exceeding 7.0 mg/dL for males and 6.0 mg/dL for females [23]. To facilitate the examination of the relationship between different stages of SUA levels and low muscle mass, SUA levels were categorized into deciles (D1, D2, D3, D4, D5, D6, D7, D8, D9, D10), with the D1 group serving as the reference

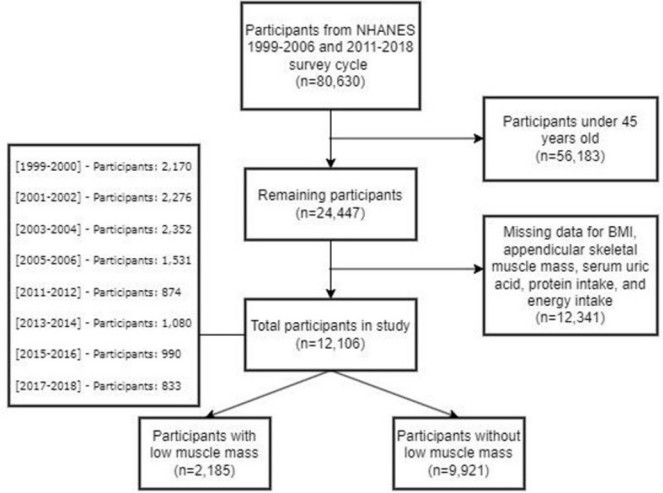

**Fig 1. Flow chart of participants.**

for analysis. It should be noted that the deciles for males and females are different due to gender-specific variations in SUA distribution (Tables 3 and 4).

## Outcome variable

The primary outcome measure is low muscle mass. Whole-body dual-energy X-ray absorptiometry (DXA) scans were conducted by the official NHANES laboratory in 1999–2006 and 2011–2018. NHANES DXA examinations provide nationally representative data on overall body composition (bone and soft tissue) and data on age, gender, and racial/ethnic groups, allowing for the exploration of associations between body composition and various health conditions and risk factors, such as cardiovascular diseases, diabetes, hypertension, physical activity, and dietary intake patterns.

According to the recommendations of the Foundation for the National Institutes of Health (FNIH) Biomarkers Consortium on Osteoarthritis, an appendicular skeletal muscle mass index (ASMI) below 0.789 for males and below 0.512 for females is considered indicative of low muscle mass. The ASMI is calculated as the total appendicular skeletal muscle mass (in kg) divided by the body mass index (kg/m$^2$) [24]. Total appendicular skeletal muscle mass is derived by summing the lean mass of the four limbs in the NHANES DXA assessments.

The DXA examinations were conducted by radiology technologists who were trained and certified. It is worth noting that the DXA conducted in NHANES 2011–2018 does not include participants older than 59 years of age. Additional details regarding the DXA examination protocol can be found in the Body Composition Procedures Manual, available on the NHANES website: https://wwwn.cdc.gov/nchs/data/nhanes/2017-2018/manuals/Body_Composition_Procedures_Manual_2018.pdf.

## Confounding variable

We adjusted our model for potential factors that may influence muscle mass based on similar past studies. Participants self-reported their age, gender, education, marital status, race, smoking status, alcohol consumption, height, weight, and income-to-poverty ratio (PIR). Alcohol status and smoking status were categorized as "yes" or "no" based on whether individuals reported consuming more than 12 alcoholic beverages annually or having smoked more than

100 cigarettes in their lifetime, respectively. Body Mass Index (BMI) was classified as under-weight/normal weight ($<25$), overweight (25–30), or obese ($>30$). Education level was categorized as below high school, high school, or above. Marital status included categories for married, living with a partner, or living alone. Race was classified as Mexican American, Other Hispanic, Non-Hispanic White, Non-Hispanic Black, Non-Hispanic Asian, or Other. Income was divided into low ($\leq1$), medium (1–3), and high ($>3$) groups based on the PIR. Dietary recall interviews were conducted prior to the Mobile Examination Center (MEC) interview to gather participants' 24-hour nutritional information, including carbohydrate and protein intake. Information on previous diseases, including diabetes, hypertension, weak/failing kidneys, coronary heart disease, angina, stroke, heart failure, and heart attack, was obtained through interviews using the Medical Conditions Questionnaire (MCQ). Given the potential impact of protein-energy wasting and comorbidities on muscle mass, we considered low energy intake ($<25$ kcal/kg/day) and low protein intake ($<0.6$ g/kg/day) [25]. Additionally, laboratory measurements included triglycerides (TG), total cholesterol (TC), blood urea nitrogen (BUN), creatinine (Cr), alanine aminotransferase (ALT), and aspartate aminotransferase (AST). Comprehensive details on specimen collection, processing, quality assurance, and monitoring are accessible on the NHANES website.

## Statistical analyses

The NHANES employed multiple imputation techniques to address missing and invalid DXA data from 1999 to 2006. Each data file includes five sets of measurements and estimates. In adherence to NHANES guidelines, we analyzed each dataset individually utilizing survey-appropriate methods and software, subsequently combining the estimates and standard errors in accordance with the combination rules delineated in the NHANES technical documentation. Appropriate period weights, stratification, and clustering were applied to amalgamate data across eight cycles from NHANES 1999–2006 and NHANES 2011–2018. For all covariates, missing values were less than 5%. We employed the 'mice' package, using random forests (RF) for categorical variables and predictive mean matching (PMM) for continuous variables, through five iterations of the multiple imputation by chained equations (MICE) method. This approach was utilized to ensure robust and unbiased estimates by handling missing data effectively, thereby enhancing the validity and reliability of our statistical analyses. All data analyses were performed using R Studio (version 4.2.1) and SAS software (version 9.4). Statistical significance was determined using a two-sided P value of less than 0.05.

Due to gender-specific differences in SUA metabolism, the data were analyzed separately for males and females. Participants were additionally categorized according to their muscle mass status, specifically whether they exhibited low muscle mass. For normally distributed variables, t-tests or chi-square tests were employed for analysis. In the case of skewed distributions, consider using non-parametric tests or Fisher's exact probability test to compare baseline characteristics among different groups. Categorical variables are expressed as weighted proportions. Continuous variables are presented as weighted means (standard error).

We employed multivariate weighted logistic regression models to explore the potential correlations among SUA, hyperuricemia, and low muscle mass, estimating OR and 95% CI. To enhance the robustness of the results, three models were analyzed. Model I was adjusted for sociodemographic characteristics, including age, race, PIR, education, and marital status. Model II, built upon Model I, included adjustments for smoking status, drinking status, BMI, low energy intake, and low protein intake. Model III, built upon Model II, additionally adjusted for various potential comorbidities including diabetes, hypertension, weak/failing

kidneys, coronary heart disease, angina, stroke, heart failure, and heart attack, as well as laboratory measurements including TG, TC, BUN, Cr, ALT, and AST. To investigate the potential nonlinear relationship between SUA and low muscle mass, restricted cubic spline (RCS) analysis with 4 knots was conducted, and the p-value for overall and nonlinearity was calculated using smooth curve fitting. Model adjustment variables in RCS analysis were consistent with Model III. Subgroup analyses were also performed, including stratification by age, BMI, PIR, education level, race, low energy intake, and low protein intake. Interactions between subgroups and SUA were examined by likelihood ratio tests.

## Results

### Participant characteristics

This study encompassed 12,106 participants aged 45 and above from eight national health and demographic survey cycles, with a low muscle mass prevalence of 22.02%. Participants were stratified by gender and the presence of low muscle mass, comprising 6,159 females and 5,947 males. Baseline characteristics of female participants revealed a sarcopenia prevalence of 20.88%. The mean age of females was 57 (9.87) years. Females with low muscle mass exhibited elevated SUA levels, a higher prevalence of hyperuricemia, older age, increased TG, elevated BUN, current alcohol consumption and a greater incidence of other comorbidities compared to those without low muscle mass (P<0.01). Except for marital status, smoking status, TC, CR, AST, and ALT, which exhibited no significant differences (P>0.05), all other variables demonstrated statistical significance (Table 1).

Table 2 delineates the baseline characteristics of male participants, indicating a low muscle mass prevalence of 23.23% with a mean age of 56.25 (9.12) years. Compared to males without low muscle mass, those with low muscle mass exhibited older age, a higher prevalence of smoking and drinking habits, elevated BUN levels, and an increased incidence of other comorbidities (P<0.01). However, SUA levels, the prevalence of hyperuricemia, marital status, and other laboratory markers did not exhibit significant differences (P>0.05).

### SUA、hyperuricemia and low muscle mass

Weighted logistic regression results, detailed in Tables 3 and 4, elucidate the association between SUA, hyperuricemia, and low muscle mass across different genders. After adjusting for all potential covariates, a positive correlation between hyperuricemia and low muscle mass was observed in females (OR, 1.43; 95% CI, 1.06 to 1.92; P = 0.021). Conversely, no significant associations were identified between either SUA levels or hyperuricemia and low muscle mass in males (P>0.05). Additionally, we conducted sensitivity analyses by stratifying SUA levels into deciles (D1-D10) to ensure the robustness of the results. Using the lowest decile (D1) as a reference and after adjusting for all covariates, no significant relationships were observed between the deciles of uric acid levels and low muscle mass in either gender (All P>0.05).

Additionally, we employed RCS adjusted for all covariates to investigate potential nonlinear associations between SUA levels and low muscle mass. For females, as depicted in Fig 2A, an increase in SUA levels exhibited a J-shaped relationship with the risk of low muscle mass, although this nonlinear relationship was not statistically significant (non-linearity p = 0.113). Conversely, Fig 2B illustrated an inverse J-shaped relationship between SUA levels and the risk of low muscle mass in males (non-linearity p = 0.638). However, smooth curve fitting indicated that the associations between SUA and low muscle mass were not statistically significant in either gender (p>0.05).

**Table 1. Characteristics of the 6159 participants with and without low muscle mass in the female population (weighted).**

| Characteristics | Female | | | |
|---|---|---|---|---|
| | Total (n = 6159) | Without low muscle mass(n = 5095) | With low muscle mass (n = 1064) | p-value |
| Age (years) | 57.00 ± 9.87 | 56.45 ± 9.55 | 61.12 ± 11.16 | <0.001** |
| Race/ethnicity, n(%) | | | | <0.001** |
| Mexican American | 5.3 | 4.0 | 15.0 | |
| Other Hispanic | 5.1 | 4.6 | 8.1 | |
| Non-Hispanic White | 73.4 | 74.4 | 66.6 | |
| Non-Hispanic Black | 10.3 | 11.2 | 4.0 | |
| Other Race | 5.9 | 5.8 | 6.3 | |
| Education level,n (%) | | | | <0.001** |
| Less than high school | 17.8 | 15.9 | 32.1 | |
| High school | 25.4 | 25.1 | 27.1 | |
| More than high school | 56.8 | 59.0 | 40.8 | |
| Marital status, n (%) | | | | 0.402 |
| Married or living with partner | 61.2 | 61.4 | 59.3 | |
| Living alone | 38.8 | 38.6 | 40.7 | |
| PIR, n (%) | | | | <0.001** |
| ≤ 1 | 11.7 | 11.0 | 16.7 | |
| 1–3 | 34.1 | 32.5 | 46.1 | |
| > 3 | 54.2 | 56.5 | 37.3 | |
| BMI (kg/m$^2$), n (%) | | | | <0.001** |
| < 25 | 31.8 | 34.8 | 10.2 | |
| 25–30 | 30.9 | 31.8 | 24.3 | |
| > 30 | 37.3 | 33.4 | 65.5 | |
| Smoking, n (%) | 44.0 | 44.6 | 39.5 | 0.051 |
| Drinking, n (%) | 62.0 | 63.9 | 47.9 | <0.001** |
| Low energy intake, n (%) | 61.3 | 59.4 | 75.4 | <0.001** |
| Low protein intake, n (%) | 25.2 | 24.2 | 33.0 | <0.001** |
| TG (mg/dl) | 145.60±100.24 | 143.56 ± 100.35 | 160.72 ± 98.12 | <0.001** |
| TC (mg/dl) | 213.22 ± 40.32 | 213.06 ± 39.90 | 214.42 ± 43.31 | 0.558 |
| BUN (mg/dl) | 13.67 ± 5.18 | 13.52 ± 4.98 | 14.79 ± 6.38 | <0.001** |
| CR (mg/dl) | 0.79 ± 0.37 | 0.80 ± 0.38 | 0.78 ± 0.27 | 0.085 |
| ALT (mg/dl) | 22.44 ± 13.21 | 22.39 ± 13.46 | 22.83 ± 11.17 | 0.440 |
| AST (mg/dl) | 24.11 ± 14.11 | 24.11 ± 14.66 | 24.13 ± 9.09 | 0.968 |
| Diabetes, n(%) | 10.6 | 9.6 | 18.3 | <0.001** |
| Hypertension, n (%) | 40.6 | 38.4 | 56.5 | <0.001** |
| Weak/failing kidneys, n (%) | 2.7 | 2.4 | 5.0 | 0.008* |
| Heart Failure, n (%) | 2.7 | 2.5 | 4.8 | <0.001** |
| Coronary heart disease, n (%) | 3.2 | 2.8 | 5.6 | <0.001** |
| Angina, n (%) | 3.6 | 3.1 | 7.5 | <0.001** |
| Heart Attack, n (%) | 3.4 | 2.9 | 7.2 | <0.001** |
| Stroke, n (%) | 3.8 | 3.4 | 6.8 | 0.001* |
| ASM (kg) | 17.67 ± 3.87 | 17.90 ± 3.86 | 15.93 ± 3.56 | <0.001** |
| ASMI | 0.62 ± 0.09 | 0.64 ± 0.08 | 0.48 ± 0.03 | <0.001** |
| SUA (mg/dl) | 4.92 ± 1.29 | 4.87 ± 1.26 | 5.31 ± 1.44 | <0.001** |
| Hyperuricemia, n (%) | 17.3 | 15.8 | 28.5 | <0.001** |

Abbreviations: SUA, serum uric acid; PIR, income-poverty ratio; BMI, body mass index; TG triglyceride; TC total cholesterol; BUN blood urea nitrogen; Cr creatinine; ALT alanine aminotransferase; AST aspartate aminotransferase; ASM, appendicular skeletal muscle mass; ASMI, appendicular skeletal muscle mass index

* p < 0.05;

**p < 0.01

**Table 2. Characteristics of the 5947 participants with and without low muscle mass in the male population (weighted).**

| Characteristics | Male | | | |
|---|---|---|---|---|
| | Total (n = 5947) | Without low muscle mass (n = 4826) | With low muscle mass (n = 1121) | p-value |
| **Age (years)** | 56.25 ± 9.12 | 55.36 ± 8.45 | 61.70 ± 10.97 | <0.001** |
| **Race/ethnicity, n(%)** | | | | <0.001** |
| Mexican American | 5.9 | 5.0 | 11.4 | |
| Other Hispanic | 4.4 | 3.9 | 7.1 | |
| Non-Hispanic White | 75.2 | 75.6 | 72.3 | |
| Non-Hispanic Black | 9.2 | 10.4 | 2.1 | |
| Other Race | 5.4 | 5.1 | 7.2 | |
| **Education level,n (%)** | | | | <0.001** |
| Less than high school | 17.2 | 15.2 | 29.3 | |
| High school | 24.6 | 24.3 | 26.8 | |
| More than high school | 58.2 | 60.5 | 44.0 | |
| **Marital status, n (%)** | | | | 0.228 |
| Married or living with partner | 74.4 | 74.9 | 71.6 | |
| Living alone | 25.6 | 25.1 | 28.4 | |
| **PIR, n (%)** | | | | <0.001** |
| ≤ 1 | 10.0 | 9.5 | 12.5 | |
| 1–3 | 29.9 | 27.2 | 45.9 | |
| > 3 | 60.1 | 63.2 | 41.6 | |
| **BMI (kg/m$^2$), n (%)** | | | | <0.001** |
| < 25 | 22.4 | 24.6 | 9.1 | |
| 25–30 | 43.5 | 45.3 | 32.1 | |
| > 30 | 34.2 | 30.1 | 58.8 | |
| **Smoking, n (%)** | 60.0 | 59.2 | 64.7 | 0.029* |
| **Drinking, n (%)** | 83.2 | 84.1 | 77.4 | <0.001** |
| **Low energy intake, n (%)** | 44.3 | 41.4 | 62.5 | <0.001** |
| **Low protein intake, n (%)** | 15.0 | 13.5 | 24.2 | <0.001** |
| **TG (mg/dl)** | 178.50 ± 165.79 | 176.40 ± 160.35 | 191.23 ± 195.19 | 0.103 |
| **TC (mg/dl)** | 203.37 ± 42.45 | 203.41 ± 41.92 | 203.13 ± 45.53 | 0.899 |
| **BUN (mg/dl)** | 14.95 ± 5.39 | 14.80 ± 5.11 | 15.89 ± 6.77 | <0.001** |
| **CR (mg/dl)** | 1.01 ± 0.48 | 1.01 ± 0.49 | 1.00 ± 0.48 | 0.401 |
| **ALT (mg/dl)** | 29.86 ± 24.43 | 30.06 ± 25.52 | 28.66 ± 16.28 | 0.170 |
| **AST (mg/dl)** | 27.47 ± 18.80 | 27.54 ± 19.49 | 27.07 ± 13.95 | 0.656 |
| **Diabetes, n(%)** | 11.9 | 10.2 | 22.3 | <0.001** |
| **Hypertension, n (%)** | 39.4 | 37.5 | 50.9 | <0.001** |
| **Weak/failing kidneys, n (%)** | 2.7 | 2.3 | 4.8 | 0.001* |
| **Heart Failure, n (%)** | 3.5 | 3.0 | 6.6 | <0.001** |
| **Coronary heart disease, n (%)** | 7.0 | 5.9 | 13.6 | <0.001** |
| **Angina, n (%)** | 5.0 | 4.0 | 10.7 | <0.001** |
| **Heart Attack, n (%)** | 6.7 | 5.5 | 14.1 | <0.001** |
| **Stroke, n (%)** | 2.7 | 2.2 | 5.8 | <0.001** |
| **ASM (kg)** | 26.34 ± 4.60 | 26.80 ± 4.42 | 23.58 ± 4.70 | <0.001** |
| **ASMI** | 0.92 ± 0.13 | 0.96 ± 0.10 | 0.73 ± 0.05 | <0.001** |
| **SUA (mg/dl)** | 6.03 ± 1.31 | 6.01 ± 1.30 | 6.13 ± 1.35 | 0.053 |
| **Hyperuricemia, n (%)** | 20.5 | 20.0 | 23.4 | 0.062 |

Abbreviations as in Table 1

* p < 0.05;

**p < 0.01

**Table 3. Weighted logistic regression to determine the odds of low muscle mass presence by hyperuricemia or SUA in the female population.**

| Variable | Non-adjusted Model | | Model I | | Model II | | Model III | |
|---|---|---|---|---|---|---|---|---|
| | OR (95%CI) | P-value | OR (95%CI) | P-value | OR (95%CI) | P-value | OR (95%CI) | P-value |
| Hyperuricemia | 2.12(1.69~2.66) | <0.001** | 1.98(1.52~2.58) | <0.001** | 1.30(0.98~1.71) | 0.065 | 1.43(1.06~1.92) | 0.021* |
| SUA, mg/dl | 1.27(1.18~1.37) | <0.001** | 1.23(1.13~1.34) | <0.001** | 1.02(0.93~1.11) | 0.674 | 1.06(0.96~1.17) | 0.253 |
| SUA, deciles | | | | | | | | |
| D1(<3.6mg/dl) | 1(Ref) | | 1(Ref) | | 1(Ref) | | 1(Ref) | |
| D2(3.6–4.0mg/dl) | 1.36(0.83~2.23) | 0.223 | 1.47(0.89~2.41) | 0.132 | 1.13(0.661~1.94) | 0.646 | 1.15(0.656~2.02) | 0.622 |
| D3(4.0–4.4mg/dl) | 1.35(0.84~2.16) | 0.214 | 1.38(0.85~2.24) | 0.192 | 1.05(0.602~1.83) | 0.865 | 1.10(0.616~1.97) | 0.741 |
| D4(4.4–4.6mg/dl) | 1.13(0.66~1.92) | 0.661 | 1.13(0.67~1.91) | 0.644 | 0.81(0.467~1.42) | 0.463 | 0.87(0.495~1.54) | 0.639 |
| D5(4.6–4.9mg/dl) | 1.16(0.70~1.94) | 0.565 | 1.17(0.70~1.96) | 0.542 | 0.74(0.430~1.29) | 0.289 | 0.78(0.444~1.40) | 0.413 |
| D6(4.9–5.3mg/dl) | 1.60(1.00~2.57) | 0.051 | 1.60(0.99~2.53) | 0.057 | 0.94(0.566~1.56) | 0.810 | 1.05(0.61~1.79) | 0.869 |
| D7(5.3–5.6mg/dl) | 1.73(1.06~2.83) | 0.029* | 1.79(1.08~2.97) | 0.025* | 0.89(0.502~1.57) | 0.674 | 0.99(0.55~1.77) | 0.969 |
| D8(5.6–6.1mg/dl) | 1.91(1.14~3.19) | 0.014 | 1.81(1.06~3.10) | 0.030 | 0.86(0.506~1.47) | 0.578 | 0.95(0.54~1.65) | 0.845 |
| D9(6.1–6.8mg/dl) | 2.84(1.80~4.48) | <0.001** | 2.93(1.83~4.71) | <0.001** | 1.30(0.768~2.20) | 0.325 | 1.46(0.84~2.55) | 0.180 |
| D10(≥6.8mg/dl) | 3.02(1.89~4.84) | <0.001** | 2.65(1.55~4.52) | <0.001** | 1.09(0.635~1.87) | 0.752 | 1.33(0.73~2.44) | 0.348 |

1 (Ref) = reference

Abbreviations: D, deciles; OR, odds ratio; CI, confidence interval; SUA, serum uric acid.

Model 1: Adjusted for age, race, PIR, education level and marital status.

Model 2: Adjusted for age, race, PIR, educational level, marital status, BMI, smoking, drinking, low energy intake and low protein intake.

Model 3: Adjusted for age, race, PIR, educational level, marital status, BMI, smoking, drinking, low energy intake, low protein intake, diabetes, hypertension, heart failure, weak/failing kidneys, coronary heart disease, angina, heart attack, stroke, TG,TC, BUN, CR, ALT and AST. The sex variables were not adjusted in the stratified analysis of sex.

* p < 0.05;

**p < 0.01

## Subgroup and sensitivity analyses

As depicted in Fig 3A and 3B, the interaction attained statistical significance exclusively within the male BMI cohort (p for interaction <0.05); no notable interaction was discerned in the other subgroups. SUA exerted a more pronounced influence on the risk of low muscle mass when BMI was below 25kg/m$^2$, likely attributable to the determination of low muscle mass by the ASMI, which is normalized for BMI. Furthermore, for the sensitivity analysis, we excluded 336 samples with missing covariate values, which corroborated the initial findings, illustrating a consistent trend in the association between SUA, hyperuricemia, and low muscle mass.

## Discussion

In this extensive cross-sectional study, the population was stratified into male and female groups due to gender differences in SUA levels and varying criteria for diagnosing hyperuricemia. After adjusting for all potential confounders, the findings reveal, for the first time, a significant positive correlation between hyperuricemia and low muscle mass among middle-aged and elderly women in the U.S., although the J-shaped association between SUA levels and low muscle mass was not statistically significant. In the male cohort, while the relationship between hyperuricemia, SUA levels, and low muscle mass was not statistically significant, an inverse J-shaped trend was observed.

This contrasts with recent studies in China, which linked higher SUA levels with better SMI and found a J-shaped relationship between SUA and grip strength in women, while reporting a negative correlation in men [20]. Another cohort study indicated a negative correlation

**Table 4. Weighted logistic regression to determine the odds of low muscle mass presence by hyperuricemia or SUA in the male population.**

| Variable | Non-adjusted Model | | Model I | | Model II | | Model III | |
|---|---|---|---|---|---|---|---|---|
| | OR (95%CI) | P-value | OR(95%CI) | P-value | OR (95%CI) | P-value | OR (95%CI) | P-value |
| Hyperuricemia | 1.22(0.99~1.51) | 0.062 | 1.20(0.95~1.52) | 0.127 | 0.90(0.70~1.15) | 0.403 | 0.91(0.70~1.17) | 0.447 |
| SUA, mg/dl | 1.07(1.00~1.15) | 0.053 | 1.09(1.01~1.17) | 0.029 | 0.95(0.88~1.03) | 0.199 | 0.96(0.89~1.04) | 0.356 |
| SUA, deciles | | | | | | | | |
| D1(<4.5mg/dl) | 1(Ref) | | 1(Ref) | | 1(Ref) | | 1(Ref) | |
| D2(4.5–5.0mg/dl) | 0.88(0.60~1.30) | 0.522 | 0.93(0.61~1.44) | 0.758 | 0.84(0.54~1.32) | 0.449 | 0.89(0.57~1.39) | 0.607 |
| D3(5.0–5.4mg/dl) | 0.83(0.56~1.21) | 0.330 | 0.89(0.60~1.31) | 0.548 | 0.70(0.47~1.04) | 0.079 | 0.78(0.52~1.17) | 0.231 |
| D4(5.4–5.7mg/dl) | 1.04(0.69~1.57) | 0.836 | 1.25(0.81~1.91) | 0.308 | 0.98(0.61~1.58) | 0.936 | 1.07(0.66~1.74) | 0.769 |
| D5(5.7–6.0mg/dl) | 0.97(0.64~1.47) | 0.890 | 1.11(0.70~1.76) | 0.647 | 0.88(0.57~1.36) | 0.561 | 0.98(0.62~1.55) | 0.945 |
| D6(6.0–6.4mg/dl) | 1.24(0.82~1.88) | 0.294 | 1.38(0.89~2.13) | 0.144 | 0.93(0.60~1.43) | 0.742 | 1.04(0.66~1.64) | 0.870 |
| D7(6.4–6.8mg/dl) | 0.88(0.58~1.35) | 0.560 | 1.13(0.73~1.73) | 0.578 | 0.77(0.49~1.21) | 0.260 | 0.88(0.56~1.38) | 0.563 |
| D8(6.8–7.2mg/dl) | 1.01(0.59~1.73) | 0.975 | 1.21(0.68~2.15) | 0.508 | 0.70(0.40~1.25) | 0.227 | 0.73(0.42~1.27) | 0.267 |
| D9(7.2–7.9mg/dl) | 1.23(0.83~1.84) | 0.298 | 1.38(0.89~2.13) | 0.153 | 0.78(0.50~1.23) | 0.286 | 0.84(0.52~1.35) | 0.470 |
| D10(≥7.9mg/dl) | 1.30(0.85~1.97) | 0.222 | 1.41(0.91~2.18) | 0.125 | 0.80(0.51~1.25) | 0.321 | 0.89(0.56~1.42) | 0.619 |

1 (Ref) = reference

Abbreviations: D, deciles; OR, odds ratio; CI, confidence interval; SUA, serum uric acid.

Model 1: Adjusted for age, race, PIR, education level and marital status.

Model 2: Adjusted for age, race, PIR, educational level, marital status, BMI, smoking, drinking, low energy intake and low protein intake.

Model 3: Adjusted for age, race, PIR, educational level, marital status, BMI, smoking, drinking, low energy intake, low protein intake, diabetes, hypertension, heart failure, weak/failing kidneys, coronary heart disease, angina, heart attack, stroke, TG,TC, BUN, CR, ALT and AST. The sex variables were not adjusted in the stratified analysis of sex.

* p < 0.05; **p < 0.01

between male SUA levels and sarcopenia, defined by gait speed, grip strength, and the five-time chair stand test, with no significant correlation observed in women [21]. Additionally, cross-sectional surveys of muscle mass or strength in various Asian populations, including

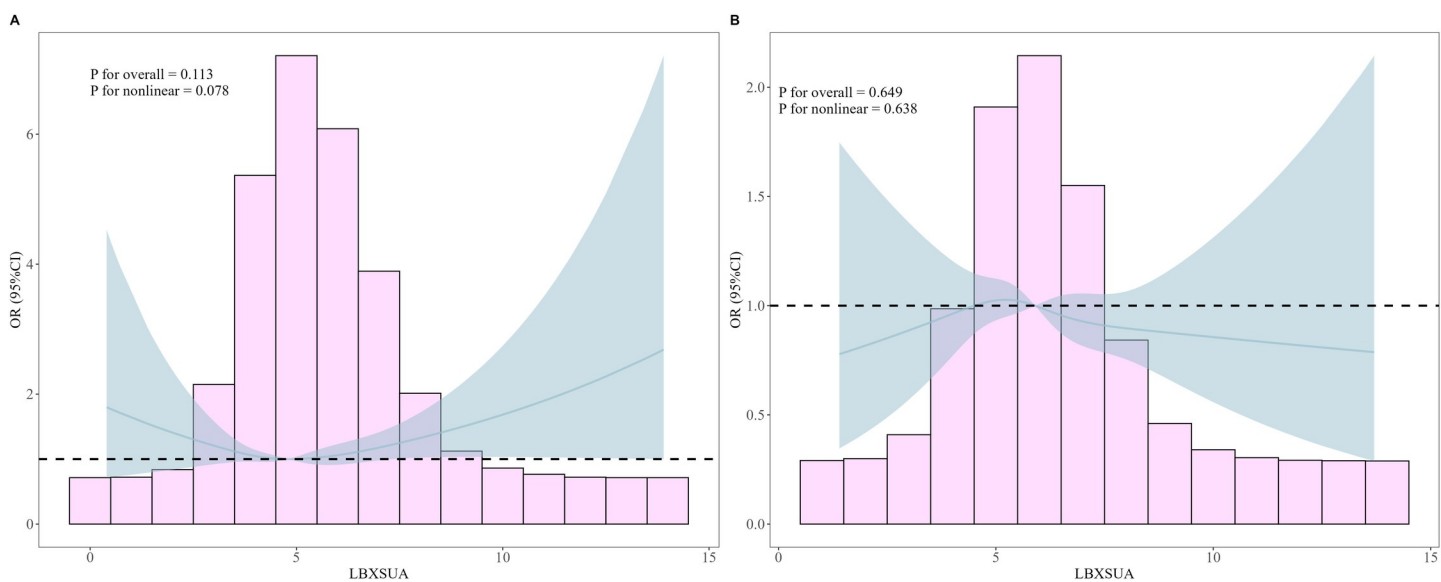

**Fig 2.** Association between SUA and low muscle mass in the female population (A) and the male population (B).

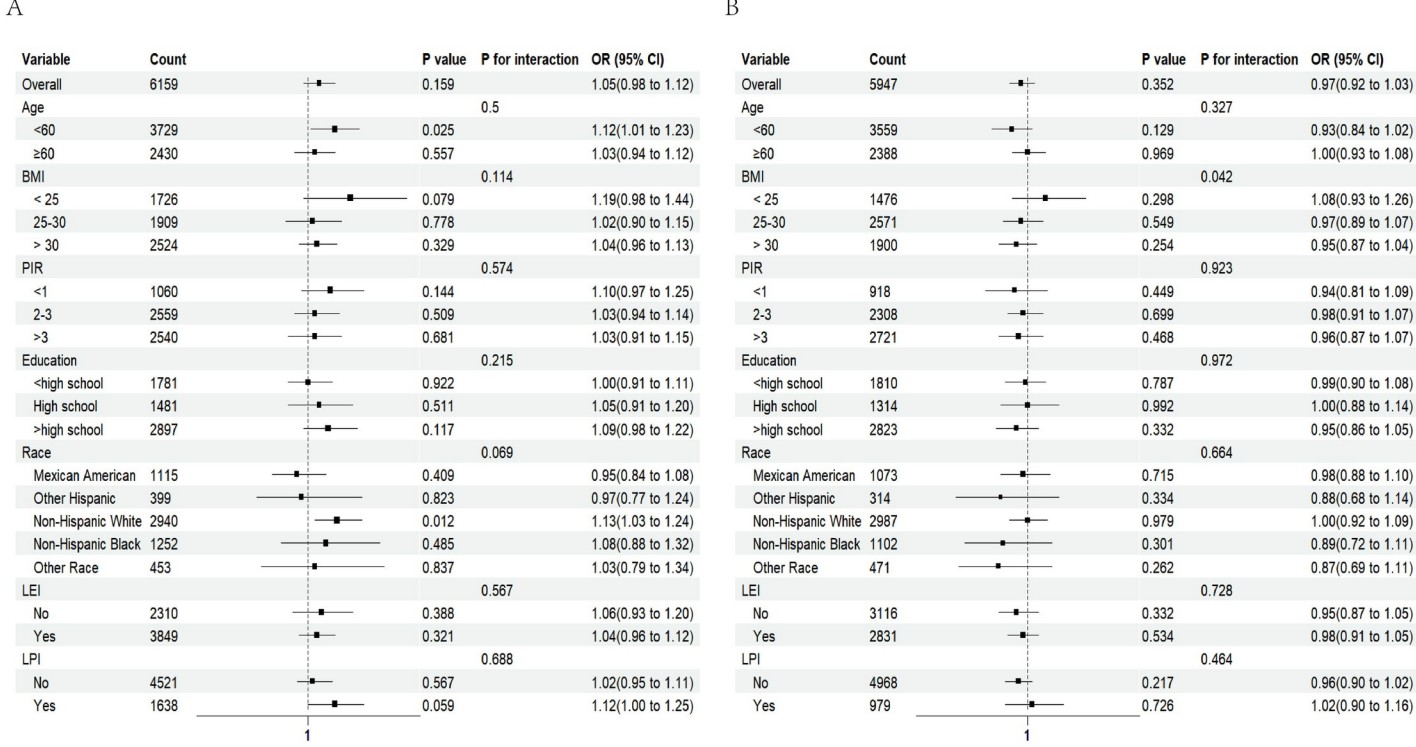

**Fig 3.** The results of subgroup analyses between SUA and low muscle mass in the female population (A) and the male population (B) after adjustment for age, race, education, marital status, PIR, alcohol status, smoking status, low energy intake, low protein intake, BMI, diabetes, hypertension, heart failure, weak/failing kidneys, coronary heart disease, angina, heart attack, stroke, TG,TC, BUN, CR, ALT and AST. Stratification variables were not adjusted in the corresponding models. LEI, low energy intake. LPI, low protein intake.

China, Japan, and Korea, have demonstrated a positive correlation between SUA levels and muscle mass or strength [16, 17, 26–28]. An Italian cohort of 239 elderly individuals also found higher SUA levels associated with better isometric grip strength [19]. In contrast to similar findings in most parts of Asia, these inconsistencies may be attributed to variations in dietary patterns, lifestyles, and genetic backgrounds across different regions. However, this also supports a Japanese study suggesting that hyperuricemia adversely affects muscle strength. The relationship between SUA levels (quartiles) and muscle strength did not follow a linear trend but exhibited an inverse J-shaped curve [14]. Additionally, this study validates the findings of a 2009 study using the same database [15], strengthening the observed relationships and expanding on the findings of Paula C Nahas and others regarding the relationship between uric acid and AMMI across a broader sample and age groups [26]. Despite ongoing ambiguities in the interplay between SUA levels and muscle mass, the primary focus of this investigation was to elucidate the relationship between uric acid and low muscle mass across different genders of middle-aged and elderly Americans. Low muscle mass, considered the primary outcome variable, was calculated using the AMMI adjusted for BMI, employing the latest classification standards for low muscle mass consistent with FNIH.

This study identified a positive correlation between susceptibility to low muscle mass and elevated SUA levels exceeding 6 mg/dL, the threshold for hyperuricemia, in the female cohort. The precise underlying mechanism governing this association remains elusive. SUA, in general, manifests robust antioxidant attributes, affording a measure of defense against oxidative cellular damage. Serving as a potent scavenger of free radicals, SUA may serve to shield skeletal

muscle functionality from protein oxidation induced by reactive oxygen species (ROS), thereby contributing to the mitigation of oxidative stress within muscular cells. This potential action holds promise for the preservation of muscular tissue health and the augmentation of muscular strength [27–30]. However, it is imperative to note that the aforementioned conclusions do not unequivocally corroborate the findings of the current study. To a certain extent, the mechanistic underpinning may emanate from the indirect escalation in ROS production linked to elevated SUA levels. ROS [31], acknowledged for its contributory role in aging processes, has been substantiated in diverse studies as an inducer of noteworthy muscle atrophy by amplifying cellular apoptosis mechanisms and impeding protein synthesis. Furthermore, the heightened levels of serum uric acid may signify an augmentation in intracellular ROS, thereby delineating a potential avenue through which this correlation operates [32–35]. Additionally, it is noteworthy that skeletal muscle can produce uric acid during metabolic alterations and periods of muscle atrophy. Consequently, the elevation in uric acid levels may not only contribute to low muscle mass but could also be a consequence of muscle atrophy [36, 37].

The free radical theory of aging has gained widespread prominence in the context of sarcopenia development [38, 39]. Uric acid has been implicated in systemic inflammation, with urate crystals serving as catalysts for an inflammatory response via the release of pro-inflammatory mediators. In this study, the risk of low muscle mass in the female cohort escalated exclusively when SUA levels surpassed 6 mg/dL, attaining the threshold for hyperuricemia. Therefore, heightened SUA levels seem to incite systemic inflammation to a certain degree, thereby augmenting the risk of sarcopenia and the consequent decline in muscle mass. Furthermore, it has been postulated that uric acid is intricately linked to various physiological processes and molecular pathways, including the production and release of nitric oxide (NO). Nitric oxide assumes a pivotal role in vasodilation and enhanced circulation within the body, potentially influencing muscle function and strength [40–42].

Gender disparities manifested in the current study. Notwithstanding the loss of statistical significance in the association between SUA levels, hyperuricemia, and low muscle mass in males following adjustments for all covariates, a discernible inverted "J-shaped" emerged. Within a specified range, as SUA levels ascended, a concomitant reduction in the risk of low muscle mass was observed, although lacking statistical significance. The origin of the observed gender disparity remains ambiguous, yet antecedent investigations suggest a plausible correlation with hormonal influences. Testosterone, the principal male sex hormone, plays a pivotal role in sustaining muscle mass by instigating muscle protein synthesis and mitigating muscle protein degradation [43, 44]. Conversely, the impact of estrogen on muscle mass in females is comparatively less pronounced than that of testosterone. Furthermore, the study cohort comprises individuals aged 45 and above, predominantly encompassing middle-aged and elderly participants. Given that a substantial proportion of the female subjects are likely to have undergone menopause, a consequential reduction in estrogen levels is probable. Literature posits that diminished estrogen levels may instigate an elevation in pro-inflammatory cytokines, such as tumor necrosis factor alpha or interleukin-6, thereby contributing to muscle catabolism [45, 46]. This mechanism may, to a certain extent, augment the correlation between SUA levels and the incidence of low muscle mass in females.

This study utilized a large, multi-ethnic sample with national representativeness to enhance the generalizability of findings to the middle-aged and elderly adult population in the United States. The NHANES officials adhered to rigorous protocols during data collection, thereby ensuring the reliability of the dataset. The large sample size further bolstered the applicability and credibility of our results. Given the metabolic disparities between genders, as well as the differential criteria for diagnosing low muscle mass and hyperuricemia, we conducted gender-

stratified analyses. Additionally, SUA levels were categorized into deciles to maintain the robustness of the outcomes. Nonetheless, this study is not without limitations. Although a large-scale cross-sectional design was employed, causal inferences regarding the relationship between SUA and low muscle mass remain constrained. Furthermore, despite referencing prior studies for covariate selection, potential residual confounding factors may have been overlooked, or the database might have lacked sufficient data for comprehensive adjustments. Lastly, due to the limitations of the database, our study solely examined muscle mass. Future research should incorporate other critical dimensions of sarcopenia, such as muscle strength and function, to provide more holistic insights. To validate our findings and extend the study population to different countries, prospective longitudinal studies are necessary.

## Conclusions

Our findings reveal a significant positive correlation between hyperuricemia and the risk of low muscle mass in middle-aged and older females in the United States, whereas the relationship between SUA levels and low muscle mass did not attain statistical significance. In the male cohort, neither SUA levels nor hyperuricemia demonstrated a significant association with low muscle mass. These observations contribute valuable insights and groundwork for further exploration of the role of uric acid in muscle mass. Nonetheless, substantiation of these associations necessitates additional longitudinal, clinical, and mechanistic studies. Such endeavors are crucial to validating these associations and offering more targeted strategies and approaches for the prevention and intervention of low muscle mass.

## Acknowledgments

We would like to express our gratitude to the NHANES (National Health and Nutrition Examination Survey) team for their extensive efforts in collecting and providing the valuable data used in this study.

## Author Contributions

**Conceptualization:** Laixi Kong.

**Data curation:** Laixi Kong, Yaqin Li.

**Formal analysis:** Yaqin Li.

**Funding acquisition:** Zhe Li, Zhenzhen Xiong.

**Investigation:** Rong Zhu.

**Methodology:** Laixi Kong.

**Resources:** Maoting Guo, Yuqing Wu, Yuxin Zhong.

**Supervision:** Zhe Li, Zhenzhen Xiong.

**Writing – original draft:** Laixi Kong, Yaqin Li.

**Writing – review & editing:** Zhe Li, Zhenzhen Xiong.

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
