## [Decision Letter · Decision Letter 0]

28 Jun 2024

PONE-D-24-13161Association between serum uric acid, hyperuricemia and sarcopenia in middle-aged and elderly adults: a national health and nutrition examination studyPLOS ONE

Dear Dr. Xiong,

Thank you for submitting your manuscript to PLOS ONE. After careful consideration, we feel that it has merit but does not fully meet PLOS ONE’s publication criteria as it currently stands. Therefore, we invite you to submit a revised version of the manuscript that addresses the points raised during the review process.

We look forward to receiving your revised manuscript.

Kind regards,

Tatsuo Shimosawa, M.D., Ph.D.

Academic Editor

PLOS ONE

Additional Editor Comments:

Three experts raised several issues to be clarified. We are looking forward to your responses.

Reviewers' comments:

Reviewer's Responses to Questions

**Comments to the Author**

1. Is the manuscript technically sound, and do the data support the conclusions?

Reviewer #1: No

Reviewer #2: Yes

Reviewer #3: Yes

2. Has the statistical analysis been performed appropriately and rigorously? 

Reviewer #1: No

Reviewer #2: I Don't Know

Reviewer #3: Yes

3. Have the authors made all data underlying the findings in their manuscript fully available?

Reviewer #1: Yes

Reviewer #2: Yes

Reviewer #3: Yes

4. Is the manuscript presented in an intelligible fashion and written in standard English?

Reviewer #1: Yes

Reviewer #2: Yes

Reviewer #3: Yes

5. Review Comments to the Author

Reviewer #1: Thank you for the opportunity to review this manuscript. Kong et al. aim to evaluate the association between uric acid and sarcopenia in a middle-aged and elderly population in the United States. While the topic is relevant, several points require clarification regarding the statistical analyses and other important factors.

Firstly, sarcopenia was defined in the study as a sarcopenia index below 0.789 for males and below 0.512 for females. This is not the correct definition of sarcopenia, which is characterized by low muscle strength in conjunction with a low appendicular muscle mass index. Therefore, the study should be revised, as the authors evaluated low muscle mass rather than sarcopenia.

Another critical point that needs clarification is the data from the DXA analysis. The authors stated, "Whole-body dual-energy X-ray absorptiometry (DXA) scans were conducted by the official NHANES laboratory in 1999-2006 and 2011-2018," and that both middle-aged and older adults were evaluated. How did the authors assess DXA data in older adults? This remains unclear, and the large number of older adults included in the study suggests potential inaccuracies. DXA data for older adults are only available until 2006, and only imputed data are available. The descriptive data indicate that the mean age was approximately 60 years, suggesting a high number of older adults in the analysis. This is problematic since DXA data for older adults are only available until 2006, while middle-aged DXA data span all the bienniums, implying that the number of middle-aged individuals should be higher than that of older adults.

Additionally, data files for 1999-2000, 2001-2002, 2003-2004, and 2005-2006 contain five sets of measured and imputed values. Each set of measured and imputed values can be merged with other NHANES data to create analytic datasets. Analysts should be aware of the highly variable nature of these imputed values when considering their use. Multiple imputation is a technique that allows analysts to incorporate the extra variability due to imputation into their analyses. Imputed values should not be treated as measured variables without accounting for this extra variability. Moreover, a single dataset should not be created using the average of the five sets of valid and imputed values. This information needs clarification before a full review of this manuscript can proceed. If the authors used the mean values of DXA data, this is incorrect and all the analyses should be redone.

The authors used several bienniums of NHANES data but did not describe the sample weight used to ensure the analyses are representative of the U.S. population. It appears that the authors may not have used the sample weight in their analyses. Please clarify whether the sample weight was used and specify which sample weight was used, given the combination of several bienniums of data.

It is very important to describe in the flow chart the number of individuals evaluated in each biennium for a correct understanding of the data.

Reviewer #2: This manuscript by Kong et al. uses data collected from NHANES to explore the relationship between serum uric acid and sarcopenia, which is calculated from appendicular lean mass measures. Male/female data were analyzed separately. This seems reasonably well done and explores a potentially interesting question. This reviewer was several suggestions that may improve the utility of the study.

1) In the Abstract, it must be stated how sarcopenia was measured and defined. This is a core measure in the study and has an incredibly large impact on interpretation of the data set.

2) Also in the Abstract, the statement that “However, an inverted “J-shaped” relationship was observed with men” must be removed. The authors found NO significant relationship in this analysis of men, which is what the main conclusion should be. Of course, it is appropriate to continue to discuss this as a trend in body of the Discussion, but it cannot be a conclusion of the study.

3) Line 113. It should be stated that the deciles are different for males and females.

4) While this reviewer appreciates the attempt to mechanistically explain the relationships between SUA and sarcopenia by uric acid acting on muscle (line 262-284), this is rather one sided. There is clear evidence that UA can be produced by skeletal muscle during changes in metabolism (PMID: 8304559) or even during periods of muscle wasting (PMID: 38032735). Therefore, increased UA may not solely be causing sarcopenia, but instead may be a consequence of muscle wasting. This should be considered.

Reviewer #3: The authors conducted a clinical study of the association between serum uric acid (SUA) levels and sarcopenia diagnosed according to FNIH criteria, for which there is little evidence to date, in the participants aged 45 years and older in the NHANES, which has issued evidence on many lifestyle-related diseases. The results showed that SUA levels and sarcopenia were significantly associated with J-Shape in women, but not in men. However, there was a trend towards an inverted J-Shape association. The study was conducted in the US, and as sarcopenia diagnosed according to FINH criteria is more common in obese individuals, it can be said that the study looked at the association between sarcopenic obesity and SUA levels, rather than sarcopenia. The reviewer has no objection to the findings of this study itself, but several points need to be confirmed.

1. The reviewer speculates that the SUA level may have affected muscle strength or physical function rather than appendicular skeletal muscle mass (ASM) in each sarcopenia indices, as women do not show a greater change in muscle mass with age than men. The authors did not show the data of each sarcopenia indices, but what would the results be if analyzed by ASM, muscle strength, and physical function, not by diagnosed sarcopenia?

2. As mentioned above, the reviewer considers that the present study examined the relationship between sarcopenic obesity and SUA levels rather than the relationship between sarcopenia and SUA levels. Therefore, the reviewer recommends that the description of sarcopenia in the title be changed to sarcopenic obesity.

6. PLOS authors have the option to publish the peer review history of their article (what does this mean?). If published, this will include your full peer review and any attached files.

Reviewer #1: No

Reviewer #2: No

Reviewer #3: No

---

## [Author Response · Author response to Decision Letter 0]

22 Jul 2024

Response to Reviewers

Response to Reviewer 1

We thank the reviewers for their detailed review of our manuscript and valuable suggestions. Firstly we have re-analysed the study data to answer the questions you mentioned and have revised and clarified the manuscript accordingly:

1. Firstly, sarcopenia was defined in the study as a sarcopenia index below 0.789 for males and below 0.512 for females. This is not the correct definition of sarcopenia, which is characterized by low muscle strength in conjunction with a low appendicular muscle mass index. Therefore, the study should be revised, as the authors evaluated low muscle mass rather than sarcopenia.

Response: Thank you for highlighting this point. We have revised the manuscript by changing the outcome variable to low muscle mass. Additionally, we have adjusted our analysis to reflect this corrected definition and updated the relevant sections of the methods and results accordingly.(line 124-139)

2. Another critical point that needs clarification is the data from the DXA analysis. The authors stated, "Whole-body dual-energy X-ray absorptiometry (DXA) scans were conducted by the official NHANES laboratory in 1999-2006 and 2011-2018," and that both middle-aged and older adults were evaluated. How did the authors assess DXA data in older adults? This remains unclear, and the large number of older adults included in the study suggests potential inaccuracies. DXA data for older adults are only available until 2006, and only imputed data are available. The descriptive data indicate that the mean age was approximately 60 years, suggesting a high number of older adults in the analysis. This is problematic since DXA data for older adults are only available until 2006, while middle-aged DXA data span all the bienniums, implying that the number of middle-aged individuals should be higher than that of older adults.

Response: Thank you for this important observation. To clarify the issue, we reexamined the DXA data source and conducted a weighted analysis. The revised results show an average age of 57 for women and 56.25 for men, which we find acceptable. We acknowledge that DXA data for older adults is only available up to 2006, while DXA data for middle-aged individuals spans all biennial periods. Upon detailed examination, we found that older adults over 60 were overrepresented and older in the four biennial cycles before 2006. This aligns with other studies (PMC10826600). We believe this demographic distribution explains why the average age of the study population across all biennial periods is approximately 57 years. In the manuscript, we also emphasized the varying ages at which DXA data were collected across different periods.(line 136-137)

3. Additionally, data files for 1999-2000, 2001-2002, 2003-2004, and 2005-2006 contain five sets of measured and imputed values. Each set of measured and imputed values can be merged with other NHANES data to create analytic datasets. Analysts should be aware of the highly variable nature of these imputed values when considering their use. Multiple imputation is a technique that allows analysts to incorporate the extra variability due to imputation into their analyses. Imputed values should not be treated as measured variables without accounting for this extra variability. Moreover, a single dataset should not be created using the average of the five sets of valid and imputed values. This information needs clarification before a full review of this manuscript can proceed. If the authors used the mean values of DXA data, this is incorrect and all the analyses should be redone.

Response: Thank you for emphasizing this issue. We confirm that in our reanalysis, we appropriately used multiple imputation techniques rather than simple mean imputation. We have included a detailed description of the multiple imputation process in the "Methods" section, specifying how the imputed values were integrated into our analysis to account for additional variability.(line 160-171)

4.The authors used several bienniums of NHANES data but did not describe the sample weight used to ensure the analyses are representative of the U.S. population. It appears that the authors may not have used the sample weight in their analyses. Please clarify whether the sample weight was used and specify which sample weight was used, given the combination of several bienniums of data.

Response: We appreciate the reviewer's suggestion. We have clarified in the manuscript that we used appropriate sample weights to ensure the analyses are representative of the U.S. population. The Methods section now includes a detailed description of the sample weights used for each biennium and how these weights were applied in the combined analysis.(line 160-171, 176-177,178)

5.It is very important to describe in the flow chart the number of individuals evaluated in each biennium for a correct understanding of the data.

Response: We have added a detailed flow chart to the manuscript that describes the number of individuals evaluated in each biennium (Fig 1). This addition is intended to provide a clear understanding of the data distribution and the analytical process.

In conclusion, we have carefully considered your thoughtful suggestions and conducted a rigorous re-analysis of our data. While the specific results have been updated to reflect this enhanced analysis, the overall findings and conclusions remain broadly consistent with our original submission. Furthermore, we have meticulously reviewed the manuscript for clarity and accuracy, addressing any language errors and ensuring the figures and tables are presented in a clear and informative manner. We believe these revisions significantly strengthen the manuscript and enhance its suitability for publication.

Response to Reviewer 2

1. In the Abstract, it must be stated how sarcopenia was measured and defined. This is a core measure in the study and has an incredibly large impact on the interpretation of the data set.

Response: Thank you for highlighting this point. In response to the other reviewers' comments, we have changed the outcome variable to low muscle mass and revised the abstract to include a clear definition of how low muscle mass is measured and defined in our study.(line 30-32, 130-132)

2. Also in the Abstract, the statement that “However, an inverted “J-shaped” relationship was observed with men” must be removed. The authors found NO significant relationship in this analysis of men, which is what the main conclusion should be. Of course, it is appropriate to continue to discuss this as a trend in body of the Discussion, but it cannot be a conclusion of the study.

Response: We appreciate the reviewer's guidance. We have removed the statement regarding the "inverted ‘J-shaped’ relationship" from the Abstract. The Abstract now correctly reflects that no significant relationship was found in the analysis of men. We have maintained the discussion of this observed trend in the Discussion section, as suggested.(line 39-45)

3. Line 113. It should be stated that the deciles are different for males and females.

Response: We have clarified in the manuscript that the deciles used in the analysis are different for males and females. This is now explicitly stated in the relevant section to ensure clarity.(line120-122)

4. While this reviewer appreciates the attempt to mechanistically explain the relationships between SUA and sarcopenia by uric acid acting on muscle (line 262-284), this is rather one-sided. There is clear evidence that UA can be produced by skeletal muscle during changes in metabolism (PMID: 8304559) or even during periods of muscle wasting (PMID: 38032735). Therefore, increased UA may not solely be causing sarcopenia, but instead may be a consequence of muscle wasting. This should be considered.

Response: We appreciate the reviewer's insightful comments regarding the bidirectional relationship between uric acid and sarcopenia. We have revised the Discussion section to incorporate the possibility that increased uric acid may not only be a cause but also a consequence of muscle wasting. We have included references to the studies (PMID: 8304559 and PMID: 38032735) to support this perspective, providing a more balanced view of the potential mechanisms involved.(line 315-317)

In conclusion, we have carefully considered your thoughtful suggestions and conducted a rigorous re-analysis of our data. While the specific results have been updated to reflect this enhanced analysis, the overall findings and conclusions remain broadly consistent with our original submission. Furthermore, we have meticulously reviewed the manuscript for clarity and accuracy, addressing any language errors and ensuring the figures and tables are presented in a clear and informative manner. We believe these revisions significantly strengthen the manuscript and enhance its suitability for publication.

Response to Reviewer 3

We sincerely appreciate your valuable feedback and insightful comments on our manuscript. We agree with your assessment that the relationship between serum uric acid (SUA) levels and sarcopenia, particularly when defined by FNIH criteria, is an area needing further investigation. We acknowledge the points you've raised and have revised our manuscript accordingly.

1.The reviewer speculates that the SUA level may have affected muscle strength or physical function rather than appendicular skeletal muscle mass (ASM) in each sarcopenia indices, as women do not show a greater change in muscle mass with age than men. The authors did not show the data of each sarcopenia indices, but what would the results be if analyzed by ASM, muscle strength, and physical function, not by diagnosed sarcopenia?

Response: We concur that SUA's influence on muscle strength and physical function, beyond just ASM, is a crucial aspect to explore. You are right to point out that women typically exhibit less age-related muscle mass loss compared to men. However, the NHANES dataset does not include comprehensive measures of muscle strength (e.g., grip strength) or physical function for the time periods included in our study. This limitation prevented us from conducting analyses using these specific parameters. We have added a statement in the limitations section of our discussion to acknowledge this.(line 

Additionally, we recognized that defining sarcopenia solely based on ASMI is insufficient. Therefore, we have revised the primary outcome variable in this study to low muscle mass, defined by ASMI according to FNIH standards. Previous research examining the relationship between ASMI and SUA found no significant association [DOI: 10.1016/j.clnesp.2022.08.034]. Consequently, this study delves deeper by investigating low muscle mass defined by ASMI across different genders, yielding meaningful results.

2.As mentioned above, the reviewer considers that the present study examined the relationship between sarcopenic obesity and SUA levels rather than the relationship between sarcopenia and SUA levels. Therefore, the reviewer recommends that the description of sarcopenia in the title be changed to sarcopenic obesity.

Response: You raise a very pertinent point regarding the prevalence of sarcopenic obesity and its potential overlap with our findings. We agree that the FNIH criteria, by using BMI-adjusted ASM, might inadvertently capture individuals with sarcopenic obesity rather than solely sarcopenia. While our data do not allow us to definitively distinguish between these groups, we have carefully revised the manuscript to acknowledge this limitation. We have also toned down the emphasis on "sarcopenia" and now focus more broadly on "low muscle mass," which is directly measured by our chosen metric.

Regarding the title change, while we understand your suggestion, we believe that changing "sarcopenia" to "sarcopenic obesity" might be overly specific. Our study primarily focuses on low muscle mass as a crucial aspect of sarcopenia. We have revised the title and other sections of the manuscript to clarify our focus on low muscle mass.

In conclusion, we have carefully considered your thoughtful suggestions and conducted a rigorous re-analysis of our data. While the specific results have been updated to reflect this enhanced analysis, the overall findings and conclusions remain broadly consistent with our original submission. Furthermore, we have meticulously reviewed the manuscript for clarity and accuracy, addressing any language errors and ensuring the figures and tables are presented in a clear and informative manner. We believe these revisions significantly strengthen the manuscript and enhance its suitability for publication.

---

## [Decision Letter · Decision Letter 1]

4 Oct 2024

Association between serum uric acid, hyperuricemia and low muscle mass in middle-aged and elderly adults: a national health and nutrition examination study

PONE-D-24-13161R1

Dear Dr. Xiong,

We’re pleased to inform you that your manuscript has been judged scientifically suitable for publication and will be formally accepted for publication once it meets all outstanding technical requirements.

Kind regards,

Tatsuo Shimosawa, M.D., Ph.D.

Academic Editor

PLOS ONE

Additional Editor Comments (optional):

Reviewers' comments:

Reviewer's Responses to Questions

**Comments to the Author**

1. If the authors have adequately addressed your comments raised in a previous round of review and you feel that this manuscript is now acceptable for publication, you may indicate that here to bypass the “Comments to the Author” section, enter your conflict of interest statement in the “Confidential to Editor” section, and submit your "Accept" recommendation.

Reviewer #2: All comments have been addressed

Reviewer #3: All comments have been addressed

2. Is the manuscript technically sound, and do the data support the conclusions?

Reviewer #2: (No Response)

Reviewer #3: Yes

3. Has the statistical analysis been performed appropriately and rigorously? 

Reviewer #2: (No Response)

Reviewer #3: Yes

4. Have the authors made all data underlying the findings in their manuscript fully available?

Reviewer #2: (No Response)

Reviewer #3: Yes

5. Is the manuscript presented in an intelligible fashion and written in standard English?

Reviewer #2: (No Response)

Reviewer #3: Yes

6. Review Comments to the Author

Reviewer #2: (No Response)

Reviewer #3: The manuscript entitled "Association between serum uric acid, hyperuricemia and low muscle mass in middle-aged and elderly adults: a national health and nutrition examination study" was revised according to the reviewer's comments and the reviewer understood and agreed with the author's intentions. There is no further comment from the reviewer.

7. PLOS authors have the option to publish the peer review history of their article (what does this mean?). If published, this will include your full peer review and any attached files.

Reviewer #2: No

Reviewer #3: No

---

## [Editor Report · Acceptance letter]

29 Oct 2024

PONE-D-24-13161R1 

PLOS ONE

Dear Dr. Xiong, 

I'm pleased to inform you that your manuscript has been deemed suitable for publication in PLOS ONE. Congratulations! Your manuscript is now being handed over to our production team.

Kind regards, 

on behalf of

Prof. Tatsuo Shimosawa 

Academic Editor

PLOS ONE